# Exploring Phenolic Compounds in Crop By-Products for Cosmetic Efficacy

**DOI:** 10.3390/ijms25115884

**Published:** 2024-05-28

**Authors:** Maria Gomez-Molina, Lorena Albaladejo-Marico, Lucia Yepes-Molina, Juan Nicolas-Espinosa, Eloy Navarro-León, Paula Garcia-Ibañez, Micaela Carvajal

**Affiliations:** 1Aquaporins Group, Centro de Edafologia y Biologia Aplicada del Segura (CEBAS-CSIC), Campus Universitario de Espinardo—25, E-30100 Murcia, Spain; mgmolina@cebas.csic.es (M.G.-M.); lalbaladejo@cebas.csic.es (L.A.-M.); lyepes@cebas.csic.es (L.Y.-M.); jnicolas@cebas.csic.es (J.N.-E.); pgibanez@cebas.csic.es (P.G.-I.); 2Department of Plant Physiology, Faculty of Sciences, University of Granada, E-18071 Granada, Spain; enleon@ugr.es

**Keywords:** secondary metabolites, phenolic, skin, cosmetic, antioxidant

## Abstract

Phenolic compounds represent a group of secondary metabolites that serve essential functions in plants. Beyond their positive impact on plants, these phenolic metabolites, often referred to as polyphenols, possess a range of biological properties that can promote skin health. Scientific research indicates that topically using phenolics derived from plants can be advantageous, but their activity and stability highly depend on storage of the source material and the extraction method. These compounds have the ability to relieve symptoms and hinder the progression of different skin diseases. Because they come from natural sources and have minimal toxicity, phenolic compounds show potential in addressing the causes and effects of skin aging, skin diseases, and various types of skin damage, such as wounds and burns. Hence, this review provides extensive information on the particular crops from which by-product phenolic compounds can be sourced, also emphasizing the need to conduct research according to proper plant material storage practices and the choice of the best extracting method, along with an examination of their specific functions and the mechanisms by which they act to protect skin.

## 1. Introduction

Some plant by-products are rich in phenolic compounds, which have been described as effective agents for health, acting against environmental oxidative stressors [1]. In this way, there is increasing attention devoted to the development of cosmetic formulations that allow for higher stability and bioavailability of bioactive phenols [2]. Phenolic compounds stand out as a crucial category of plant secondary metabolites, playing a significant role in morphological development, physiological processes, and reproduction. Their synthesis occurs through the pentose phosphate, shikimate, and phenylpropanoid pathways. The diverse biological properties of phenolics, which are attributed to their molecular structure, are well recognized. The structure of the majority of phenolic compounds comprises at least one phenol ring, typically with hydrogen replaced by a more reactive residue like hydroxyl, methyl, or acetyl [3].

In this review, we establish the relationship between external application of plant phenolic compounds and human skin health. The contributions reviewed open new perspectives towards the exploitation of phenol-rich natural extracts obtained from plant by-products as functional ingredients in the cosmetic sector. This consideration includes typical Mediterranean crops and highlights the gaps in knowledge that need to be addressed for the integration of these extracts into the cosmetic sector.

## 2. Composition in Phenolic Compounds of By-Products

A great diversity of phenolic compounds is found in the by-products of various crops such as artichoke, broccoli, cauliflower, citrus, grape, tomato, onion, and mushroom. Phenolic compounds can be extracted from different parts of plants, including fruits, leaves, stems, roots, seeds, and flowers, and often, these parts are considered by-products or waste in the food industry and agriculture [4,5,6,7]. The type and content of these phenolic compounds in by-products depend on several factors, such as plant species, cultivar, maturity stage, processing method, storage condition, and extraction technique [6,8,9]. The main phenolic compounds found in some crop by-products are described below and summarized in Table 1.

Artichoke (*Cynara scolymus* L.) is a vegetable commonly found in Mediterranean areas, whose edible parts are the immature inflorescences called heads or capitula. To prepare fresh, canned, or frozen artichoke products, the outer parts, stems, and bracts are removed, generating these by-products that comprise 70% to 80% of the total artichoke head. However, a significant portion of the plant, including leaves and stems, is discarded during industrial processing despite containing valuable bioactive compounds such as phenolic acids and flavonoids [6,10]. Chlorogenic and cynarin acids are the predominant phenolic compounds found in artichoke, while luteolin and apigenin derivatives are normally present [11,12,13]. The content and composition of these compounds vary depending on the part of the plant. Thus, the highest concentration of phenolic compounds is typically found in the parts closest to the artichoke heart [12]. Some phenolic compounds such as caffeoylquinic, quinic, and chlorogenic acids are presented in all waste parts, but neochlorogenic and cryptochlorogenic acids are only found in bracts and receptacles [8,10,13].

Other vegetables such as the *Brassica oleracea* species, whose inflorescence is consumed as broccoli and cauliflower, contain high concentrations of phenolic compounds in all aboveground parts, including leaves and stalks, which make up 70% of the aerial biomass [14,15]. Hydroxycinnamic acids are the main phenolic compounds found (more than 92% of total phenols), but neochlorogenic acids and chlorogenic acids are also highly found in these species [14,16,17]. Specifically, broccoli by-products have a high concentration of kaempferol, quercetin, and caffeic acid [17,18]. The phenol content varies according to the part of the plant, with a higher content of total phenols and specifically flavonoids in the leaves than in the florets, stalks, and cores [18,19] but it also varies between different cultivars [19].

Other types of crops that contain a huge diversity of phenolics compounds are fruit trees such as *Citrus* sp. This genus includes orange, tangerine, lemon, and grapefruits, which are consumed as fresh fruit or processed into juice [6]. Millions of tons of citrus wastes are generated annually, which can be a valuable source of polyphenols and flavonoids [20]. The by-products generated from citrus processing, including the peel (flavedo and albedo), pulp residue (rag), and seeds, make up over 50% of the fruit. These by-products are mainly rich in flavonoids [6]. Accordingly, the peels of citrus fruits, particularly oranges, have higher phenolic content compared to the edible portions, containing compounds like hesperidin. The water extracts of citrus pomace also have significant phenolic and flavonoid content [21], with high contents of hesperetin-7-*O*-rutinoside, quercetin, peonidin, and apigenin [20,21,22,23].

Grape (*Vitis vinifera* L.) is another fruit of great economic importance, with Europe being the leading wine producer globally and France, Italy, and Spain as the top countries. The winemaking process generates significant amounts of by-products, such as vine shoots, grape stalks, wine lees, and grape pomace. In this way, grape pomace, which consists of peels, pulp, and seeds, accounts for 62% of the organic waste [6] and is very rich in phenolic compounds, particularly quercetin [4,24]. But it has also been described that grape pomace extraction shows high diversity of anthocyanins such as malvidin [24,25]. Additionally, grape by-products are also the best source of resveratrol, a stilbenoid type of natural phenol with great antioxidant characteristics [26]. In grape, the variety factor is very important, as different grape varieties exhibit specific distributions of phenolic compounds [27].

The cultivation and processing of tomato (*Solanum lycopersicum* L.) is also among the most waste-producing crops and with a higher content of phenolic compounds. Tomato waste residues include rotten ripe or defective tomatoes, stems, leaves, branches, and tomato fruit processing wastes such as pulp, seeds, and peels [6,28]. The main polyphenols found in tomato by-products are flavonoids, with naringenin and chrysin being the predominant compounds [16,29,30]. Tomato peels contain a higher concentration of phenolic compounds compared to seeds and pulp, with flavonols such as quercetin, naringenin, and rutin [7]. The phenolic content tends to increase during the stages of maturation, but genetic control and environmental factors also influence the accumulation of polyphenols [28]. Tomato pomace, which is the remaining solid waste from the industry, has a lower content of phenolic compounds compared to other by-products as grape, but it still contains phenolic acids and flavonoids such as naringenin and naringenin chalcone [4].

In the case of onion (*Allium cepa* L.), the agroindustry generates a significant amount of waste, accounting for approximately 15% of total production [31]. The content and location of phenols in onions vary depending on factors such as the layer, color, and type of bulbs [32]. The main components of onion waste are the outer fleshy layers, top and bottom bulb parts including roots, onion skins produced during mechanical peeling, and undersized or damaged onion bulbs [33]. Onion skins, outer fleshy scales, and internal scales are rich in phenolics, including flavonoids like quercetin and its derivatives. Additionally, onion skins contain other phenolic compounds, including 4-hydroxybenzoic acid, protocatechuic acid, and vanillic acids. Specifically, quercetin-4-*O*-glucoside is the major compound found in both onion peels and skins [32,33,34].

In recent years, the cultivation of mushroom species on different substrates has been gaining importance. It was observed that this cultivation can generate by-products such as mycelia, stipes, caps, not-commercial mushrooms, and spent mushroom substrate with a considerable content of phenolic compounds [5,35] Mushrooms do not have the enzymes to synthesize flavonoids, although they can absorb these compounds from the substrate or plants from which they form mycorrhizae [36]. The profile of phenolic compounds varies according to the mycorrhizae species, but in general, gallic acid is the most predominant in fermentates and dry powder produced from residues derived from the cultivation of various mushroom species together with chlorogenic, cinnamic, coumaric, and 4-hydroxybenzoic acids [5,37,38,39].

The extraordinary composition of the plant material by-products should be considered as primary material. As it can be observed by the concentration variation reported in the by-product collection (Table 1), type and time should be the important factors for the recovery of important secondary metabolites. Therefore, harvest and storage should be carried out in the same way as for food material, or all the bioactive compounds will be lost.
ijms-25-05884-t001_Table 1Table 1Main phenolic compounds found in different crop by-products. The compounds appear in alphabetic order. Those in bold have been referenced with bioactivity in the cosmetic industry.SpeciesPhenolic CompoundsConcentration (mg g^−1^ DW)ReferencesArtichokeApigenin, **Caffeic acid**, Caffeoylquinic acid, Chlorogenic acid, Coumaric acid, Cynarin, **Ferulic acid**, Luteolin, Naringenin, Narirutin, Quinic acid1.99–27.21[8,10,11,12]Broccoli**Caffeic acid**, Chlorogenic acid, Coumaroylquinic acid, **Ferulic acid**, Kaempferol, **Quercetin**, Sinapic acid6.70–163.18[14,16,18,19]Cauliflower**Caffeic acid**, Coumaric acid, **Ferulic acid**, Kaempferol, Lutein, **Quercetin**, Sinapic acid3.45–10.70[17,40,41]*Citrus* sp.Apigenin, Benzoic acid, Caffeic acid, Caffeine, Caffeoylquinic acid, Caffeoyltartaric acid, Catechol, **Catechin**, Chlorogenic acid, Cinnamic acid, Cyanidin, Delphinidin, Dihydroxyflavone, Diosmin, Ellagic acid, Eriocitrin, Eriodictyol, Ferulic acid, **Gallic acid**, Gentisic acid, Hesperetin, Isorhamnetin, Isosakuranetin, Kaempferol, Luteolin, Malvidin, Myricetin, Naringenin, Naringin, Narirutin, Neoeriocitrin, Neohesperidin, 4-hydroxybenzoic acid, Pelargonidin, Peonidin, Phloretin, Propylgallate, **Quercetin**, **Resveratrol**, **Rosmarinic acid**, Rutin, Scopoletin, Sinapic acid, Sinensetin, Syringic acid, Vanillin, Vanillic acid, Vitexin3.00–196.20[20,21,22,23,30,42,43,44,45]GrapeApigenin, Catechol, **Catechin**, Caftaric acid, Chlorogenic acid, Chrysoeriol, Cinnamic acid, Coumaric acid, Coutaric acid, Cyanidin, Delphinidin, Ellagic acid, Epicatechin, Eriodictyol, **Ferulic acid**, **Gallic acid**, Hesperidin, Hydroxybenzoic acid, Hyperoside, Isorhammnetin, Kaempferol, Luteolin, Malvidin, Morin, Myricetin, Naringenin, Peonidin, Petunidin, Procyanidin, Protocatechuic acid, Pyrogallol, **Quercetin, Resveratrol**, Rosmarinic acid, Rutin, Sinapaldehyde, Synergistic acid, Syringic acid, Vanillic acid, ε-viniferin2.05–110.00[4,24,27,46,47,48,49,50]TomatoApigenin, **Caffeic acid**, **Catechin**, Chloretic acid, Chlorogenic acid, Chrysin, Cinnamic acid, Coumaric acid, Eugenol, **Gallic acid**, Isoquercetin, Isorhamnetin, Kaempferol, Luteolin, Myricetin, Naringenin, 4-hydroxybenzoic acid, Protocatechuic acid, **Quercetin, Resveratrol**, Rutin, Sinapic acid, Syringic acid, Vanillic acid0.39–9.45[4,16,24,29,30,51,52]OnionCoumaric acid, **Ferulic acid**, Isorhamnetin, Kaempferol, Myricetin, 4-hydroxybenzoic acid, Protocatechuic acid, **Quercetin**, Vanillic acid, Vanillinic acid16.90–100.00[32,34]Mushrooms**Caffeic acid**, **Catechin**, Catechin gallate, Chlorogenic acid, Cinnamic acid, Coumaric acid, **Ferulic acid, Gallic acid**, Luteolin, Myricetin, Naringenin, 4-hydroxybenzoic acid, Protocateic acid, Rutin, Syringic acid, Vanillic acid1.84–12.00[5,53,54,55,56]


## 3. Extraction Processes and Inner Stability of Compounds

Conventional extraction including solid–liquid extraction (SLE) or Soxhlet extraction, liquid–liquid extraction (LLE), and maceration are the main methods used for phenolic extraction that depend on type of sample (solid vs. liquid) to extract any type of phenolic compound of interest, polar or non-polar. However, new methods have been also developed in order to fulfil the niche of the conventional methods misses or flaws. Although there are many of them, the more suitable for phenolic extraction is supercritical CO_2_ extraction (SC-CO_2_). Accordingly, the specific process used will provide a different profile of phenolic compounds and should be chosen according to the need for a higher concentration of actives. We herein review some of them that offer high feasibility with plant phenolic compounds.

Conventional extraction methods have long been used for the extraction of phenolic compounds from various plant sources. These methods include maceration, decoction, percolation, infusion, digestion, serial exhaustive extraction, and Soxhlet extraction [57]. But some of them are not recommended for phenolic extraction due to the degradation of the samples.

Maceration is a simple method that involves immersing plant material in an appropriate solvent (for polar or non-polar polyphenols) within a closed system and agitating it at room temperature [58]. Once the soaking process is complete, the solid plant material needs to be separated from the solvent, which can be achieved through filtration, decantation, or clarification methods [59]. However, despite its straightforward nature, maceration has some drawbacks. It can be time-consuming and requires a significant volume of solvent, leading to higher costs. Although maceration has been used for phenolic compounds extraction of artichoke heads [60,61], *Citrus reticulata* [62], and citrus peel [63] the yield is lower than other methods.

Percolation, closely resembling maceration, involves placing finely powdered samples within a sealed system, followed by the gradual introduction of solvent from the upper to the lower regions [64]. It should be noted that the extraction equipment is commonly integrated with filters that allow solvent passage. Nevertheless, percolation shares common challenges with maceration, including extended extraction times, significant solvent volumes, and considerations related to the solubility of polyphenols, sample size, and extraction duration. Modified applications of percolation have been explored, particularly in the context of artichoke extractions [65]. However, the method is seldom used in contemporary practice, although yield highly than maceration but lower that straightforward and higher-yield alternatives.

With Soxhlet extraction process, powdered samples are placed in nitrocellulose thimbles within an extraction chamber equipped with a reflux condenser and positioned above a collecting flask. The solvent in the heating bottle is vaporized, and the resulting vapor condenses, returning to the thimbles containing the sample. Excessive heating can potentially impact the extraction of thermolabile polyphenols [66,67]. Furthermore, Antony and Farid (2022) [66] claimed that more investigation is needed to understand the behavior of polyphenols during extraction at high temperatures. These methods involve using specific organic solvents at defined concentrations, including methanol, water, chloroform, n-hexane, propanol, ethyl acetate, and acetone, each with varying polarities that influence phytochemical extraction. The use of these solvents mixed makes them suitable for enhancing extraction yields [68]. Therefore, although this Soxhlet technique has been widely used for phenolics extraction, such as in broccoli plants [69,70], citrus [71], grape [72] and artichoke [73], the choice of right solvent for specific phenolic compounds should be determined since there is no universally best solvent mix.

Most laboratories prefer conventional extraction methods due to their affordability and simplicity. However, some innovative methods have emerged in recent years, including supercritical CO_2_ extraction (SC-CO_2_). While originally tailored for non-polar compounds, SC-CO_2_ can be adapted for the extraction of polar compounds as well [74], which has been extensively used for the extraction of resveratrol from grape pomace. The most used solvent has been 5% ethanol at modified pressures of 100–400 bar and temperatures of 35–55 °C [75]. SC-CO_2_ has also been optimized for extracting phenolic compounds from broccoli leaves, resulting in high extraction yields [76]. In the same way, microwave-assisted extraction (MAE) [77] is used for artichoke [78], broccoli [69], citrus [79,80], and tomato [81]. Also, ultrasound-assisted extraction (UAE) [82] is commonly used for artichoke [83,84,85], broccoli [86], tomato [16], and grape [87]. Enzyme-assisted extraction (EAE) has garnered attention for its favorable impact on polyphenol extraction. Notably, studies have reported substantial polyphenol yields when employing cellucast^®^, pectinex^®^, and novoferm^®^ enzymes as pretreatment agents during the extraction of polyphenols from grape waste [88]. Pressurized fluid extraction (PFE) is used for extracted anthocyanins from grape skins in which a solvent mixture of HCl, acetone, methanol, and water (0.1:40:40:20) was found the most effective [89].

These unconventional methods offer advantages such as reduced solvent usage, improved yields, fewer toxic residues, better reproducibility, and, in some cases, shorter extraction times. These methods can also be combined for enhanced extraction efficiency [90]. Therefore, parameters such as solvent type, solvent-to-sample ratio, extraction time, temperature, and pressure should be always characterized to enhance the extraction efficiency and yield of phenolic compounds from plant material. Therefore, while all of these extraction methods have the potential to yield high amounts of phenolic compounds, the actual yield mostly depends on the plant material. Hence, it is essential to optimize extraction conditions for each method to achieve the highest possible yield from a given sample. Additionally, a combination of different extraction techniques or the use of sequential extraction steps may be very useful to maximize the overall yield of phenolic compounds.

## 4. Cosmetic Formulation

Due to the high competition in the cosmetic market and the incessant demand for new and increasingly natural products, phenolic compounds have received special attention in recent years, as they offer numerous benefits for the skin and hair [91]. In addition, the beneficial properties in cosmetics of phenolics compounds, such as whitening, anti-aging, moisturizing, or antioxidant activities, have been the subject of study recently [92].

The concentrations of phenols in cosmetic products can vary widely depending on the type of phenolic compound, the product format, its specific function, and the regulations of the country where the product is marketed. In the European Union (EU), a high number of cosmetics containing phenols in their formulations are commercially available (Figure 1). However, some of these formulations are under patent, making it impossible to know their exact composition and concentrations.

When formulating, particularly with phenolic compounds, there are several physicochemical characteristics that must be considered. Emulsified systems, also known as emulsions, are the most common and widely known formulation in the cosmetic industry [93]. They can be defined as the mixture of two immiscible liquids, such as oil and water, which remain stable due to the presence of an emulsifying agent. Nevertheless, detrimental phenomena like lipid oxidation might occur at the interphase of the emulsion. This is usually mainly provoked by the presence of water-soluble pro-oxidants and the degree of unsaturation of the lipid phase [94]. In this way, phenolic compounds (e.g., flavonols, caffeic acid, etc.) are a great option to include in these formulas since their action as antioxidants can both prevent the lipid oxidation and act as a bioactive. However, the optimal concentration of phenolic compounds present in the formula can be affected by their oxidation and subsequent deactivation. Consequently, there are different parameters that must be considered in order to formulate with phenolic compounds in order to achieve a stable emulsion with an optimal concentration of bioactives that serve both purposes: increasing the preservation of the formula and contributing to the effectiveness of the cosmetic.

The physicochemical properties of the molecules, such as their polarity, can directly affect diverse properties like solubility, reactivity, and their capacity to react with other compounds or molecules. The polarity is determined by the oil–water partition coefficient ($\log P_{W}^{O}$), which can be defined as the logarithm of the ratio of phenolic compound concentration present in the oil phase to the water phase without the presence of emulsifier. Thus, a negative partition coefficient value corresponds with a higher presence of the phenolic compound in the water part of the system, meanwhile a positive value indicates a higher proportion of it in the oil phase [95].

In an emulsion system, there are three distinct phases: the aqueous phase, the oil phase, and the oil–water interface (Figure 2). The distribution of the phenolic compounds in these phases will depend on their nature. However, it is difficult to define the exact distribution of a phenolic compound in the different phases of the emulsion system based only on the polarity. Since the oil–water interface is a narrow and anisotropic region surrounding the emulsion droplets, its structural composition is directly influenced by the type and concentration of the molecules present in it. In this way, the thermodynamic features of the phenolic compounds must also be considered in order to comprehend their adsorption to the surface of the droplet [96].

Below are the main polar and non-polar phenolic compounds of interest for their bioactivity in cosmetics, whose structure can be observed in Figure 3.

### 4.1. Polar Phenolic Compounds

Caffeic acid, i.e., 3,4-dihydroxycinnamic acid, is mainly found in the vegetable sources reviewed in this manuscript, such as artichoke, broccoli, cauliflower, tomato and mushrooms. Resonance and conjugation effects occur in the caffeic acid molecule due to the presence of the CH=CH bridge between the carboxyl group and the aromatic ring [97]. Thus, its antioxidative effect in oil-in-water emulsions is directly correlated with its ability to delay the initiation and propagation stages of lipid oxidation by donating an H atom to free radicals or by acting as a metal chelator [98].

Caffeic acid and chlorogenic acid are two hydroxycinnamic acids recognized for their potent antioxidant capabilities, which grant them antibacterial and antiviral properties. Moreover, they enhance skin texture by illuminating it and combatting signs of aging. They have the ability to increase collagen production and protect the epidermis from UV rays [99]. Both of these acids can be found in concentrated serums from various brands such as The Ordinary, SkinBetter, and Skinphysics.

Gallic acid, i.e., 3,4,5-trihydroxybenzoic acid, has been included in different formulas thanks to its ability to accept electrons and hold charges due to the presence of hydroxyl groups in an ortho-position. This gives the molecule a coplanar and bent configuration that allows its antioxidant activity [100]. In addition, gallic acid is mainly found in red grapes, berries, diverse nuts, and tea-derived extracts. In an emulsifying system, gallic acid has demonstrated its ability to increase the interfacial area of an oil-in-water emulsion in a dose-dependent trend [101]. This highly contributes to decreasing the emulsion since it is a negatively charged molecule, preventing the droplets from interacting and precipitating.

Catechin is a polyphenolic compound commonly present in extracts derived from green tea, berries, cocoa, or grape seeds. Also known as flav-3-ol, catechin has proven to be competitively adsorbed in the oil–water interface, decreasing the interfacial tension in a concentration-dependent way when introduced in an O/W emulsion. This characteristic is due to the presence of a central heterocyclic oxygenated ring and two benzene rings [102]. In addition, it has been reported that this structure grants catechin a high ability to interact with the emulsifiers through hydrophobic, hydrophilic, or covalent interactions, which increases its ability of stabilize the oil–water interface of the emulsion, thus giving a highly stable cosmetic emulsion [103].

Rosmarinic acid is a polyphenol mainly derived from rosemary, sage, and thyme. This molecule consists of two aromatic rings, each bearing a hydroxyl group in the ortho-positions. These groups have demonstrated the potential to donate hydrogen atoms to lipid free radicals. In this way, rosmarinic acid shows a high ability to inhibit the formation of volatiles and hydroxoperoxides. Also, rosmarinic acid has shown a high capacity for quenching superoxide anion radicals present in oil-in-water emulsions [104]. Nevertheless, it has been demonstrated that rosmarinic acid antioxidant properties are strongly affected by pH, decreasing from pH 5 to pH 7 [105].

### 4.2. Non-Polar Phenolic Compounds

Ferulic acid, also known as 4-hydroxy-methoxycinnamic acid, is a derivative of cinnamic acid that presents an elevated antioxidant capacity due to the presence of the phenolic ring and the unsaturated side chain, which can result into a resonance stabilized phenoxy radical [106]. Ferulic acid can be mainly found in extracts derived from some cereals, flaxseeds, and vegetables like pineapple, bananas, spinach, beetroot, artichokes, and coffee beans [107]. In addition, ferulic acid has shown a great stability in both oil-in-water and water-in-oil emulsions, achieving a great permeation coefficient under diverse formulas [108]. However, it has been demonstrated that ferulic acid shows no antioxidant activity in O/W emulsions, with a high proportion in ω-3 fatty acids stabilized with whey proteins [109]. According to partition coefficient of ferulic acid, in a system of fish oil–water, it was observed that its dissociated form is highly hydrophilic, but its undissociated form presents a higher hydrophobicity, which is a relevant fact to consider when formulating [110].

Resveratrol is a novel ingredient in cosmetics owing to its anti-aging activity. This compound has the ability to penetrate the skin barrier and stimulate fibroblast proliferation while also increasing collagen III concentration. Like other phenols, its antioxidant capacity enables it to shield the skin from UV radiation and mitigate the process of skin photoaging. In such products, it is often present at concentrations of around 5% either as a pure ingredient or as part of a grape extract [111].

Quercetin is mainly present in extracts derived from black tea, apples, berries, and onions and contains a 3-hydroxyflavone backbone with five hydroxyl groups located in position 3-4 and 3-5-7. Thanks to this structure, it provides three complexing sites to metal cations, acting as a strong chelating agent. Furthermore, quercetin possesses a poised equilibrium due to the presence of the catechol structure (3,4-dihydroxy in the B ring) and the 3,5-OH groups present in the C ring. This enables this molecule to strike a balance between associating with the lipid droplet surface to inhibit lipid oxidation and engaging with ferric ions (Fe^3+^) present within the aqueous phase [112].

## 5. In Vitro Effects

This section intends to summarize and analyze several in vitro assays carried out to explore the cosmetic effects of phenolic compounds. The initial phase entails establishing the effective concentration (EC50), which represents the quantity of the compound required to produce a 50% response. Thereafter, assays are performed to evaluate different features as the antioxidant effect, the anti-inflammatory capacity, or the antimicrobial properties [113].

### 5.1. Antioxidant Effect

Oxidative stress is a condition that affects the skin and can be caused by internal and external factors. Aging and exposure to solar radiation (UV-A and UV-B) are notable factors. This kind of stress primarily produces reactive oxygen species (ROS) that affect cells at various levels. Therefore, plant extracts rich in phenolic compounds have been used to prevent or treat the oxidative stress effects. Assays to evaluate the antioxidant potential are based on determining the ability to neutralize free radicals and reduce oxidative stress [114]. This is commonly done through the use of various spectrophotometric methods, including the DPPH assay (1,1-diphenyl-2-picrylhydrazyl), ferric reducing antioxidant power (FRAP) assay, and oxygen radical absorbance capacity (ORAC) assay.

Artichoke extracts, as stated above, are rich in different polyphenols, including hydroxycinnamic acids and flavonoids. Chlorogenic acid, a hydroxycinnamic acid, acts as a free radical scavenger and a UV protector [115] Studies conducted using extracts derived from artichoke have yielded noteworthy findings. Ethanol-based extracts from artichoke heads have been shown to enhance the functions of endothelial cells whilst also increasing the expression of genes responsible for oxidative stress protection and the formation of tight junctions, which play a crucial role in maintaining the structure of skin cells. Additionally, they affect other genes such as VEGF, ET-1, or eNOS, which are involved in angiogenesis and the elevation of capillary permeability—factors affected during ageing [61]. Furthermore, three types of artichoke leaf extract (infusion, decoction, and hydroalcoholic) exhibited notable scavenging capacity [116]. In addition, artichoke extract also contains compounds that act as an in vitro solar protection factor (SPF), including flavonoids [117].

Citrus species possess multiple health-promoting properties due to the presence of different bioactive compounds, such as phenolics [118]. Concerning skincare, the ability to protect and prevent UV-induced damage of extracts from red orange (*Citrus sinensis* L. *Osbeck*) was examined for their ability to protect and prevent UV-induced damage in fibroblasts and keratinocytes. It was reported that they prevent oxidative stress by averting DNA damage and extracellular matrix degradation [119]. *Citrus limon* peel extracts were also tested to protect keratinocyte cells against oxidative stress. This is achieved through the regulation of the Nrf2/HO-1 signaling pathway by improving antioxidant enzymes such as SOD, GSH, and CAT. Compounds identified in the extract, such as gallic acid, catechin, or caffeic acid, were attributed to this effect [120]. The *Brassica* genus, specifically broccoli (*B. oleracea* var. *italica*), is widely known due to its beneficial properties. Extracts enriched in phenolic compounds from various broccoli by-products, such as leaves, exhibit high antioxidant activity. This is attributed to a variety of compounds, including kaempferol [121], whose action mechanism has been partially elucidated and is linked to the ROS/JNK/NF-κB signaling pathway, demonstrating its potential as a suitable cosmetic ingredient against dermal fibroblastic inflammation and oxidative damage [122]. These properties have been previously described in various plant-derived extracts abundant in phenolic compounds. For example, *Aloe vera* by-products (skin) exhibit in vitro free radical scavenging activity due to their phytochemicals and antioxidants [123]. The wine production industry also generates significant by-products. Grape-derived by-products, with their high content of phenolics, have been tested in vitro for their antioxidant activity and potential use in the cosmetic industry [124,125,126]. Similarly, the berry industry generates agricultural waste, with leaves being a by-product that was used to make extracts, proving their significant ability to remove free radicals in keratinocytes and fibroblasts in vitro [127]. On the other hand, coffee silverskin, the predominant solid by-product from the coffee-roasting process, was used to obtain enriched phenolic extracts that showed an antioxidant activity ranging from 206 to 287 μmol Trolox equivalent/g and 95 to 217 μmol Fe^2+^/g by DPPH and FRAP methods. Moreover, the tested concentrations were not cytotoxic on keratinocytes and fibroblasts [128]. Similarly, extracts obtained from pomegranate were tested on human keratinocytes and showed no toxicity at any of the concentrations tested [129]. Another specific mode of action for these types of extract is to enhance the synthesis of collagen and elastin, crucial proteins for skin firmness and elasticity, which are affected by oxidative stress caused by UV radiation, and these can be measured by using Western blot or ELISA techniques specific to these proteins. The degradation of both collagen and elastin is crucial to the aging process. Therefore, addressing this level is also important, and it has been shown that phenolic compounds have this effect as well [130,131].

### 5.2. Anti-Inflammatory Effects

Similar to the above description regarding antioxidant activity, a multitude of in vitro assays can be used to study the anti-inflammatory effects of various plant extracts rich in antioxidant compounds. The anti-inflammatory capacity of the compounds is assessed by measuring the inhibition of inflammatory mediator production through techniques like ELISA or quantitative PCR. Additionally, oxidative stress and inflammation are physiologically interconnected, so most tests encompass a dual approach to assess both activities, as they will be related to each other. In this context, research can be found with extracts from different sources: citrus peels, which were able to inhibit the production of pro-inflammatory cytokines (TNF-α, IL-6, and IL-1β) [132]; artichoke, which were able to inhibit pro-inflammatory mediators (IL-6 and monocyte chemoattractant protein 1) [133]; formulations based on red grape pomace extract, which decreased the release of the pro-inflammatory cytokine IL-8 [134]; and a formulation based on the cell membrane component from broccoli leaf, which decreased the production of pro-inflammatory cytokines (IL-1*β*, IL-6, and TNF-*α*) in human macrophages [135].

### 5.3. Antimicrobial Effect

The antimicrobial capacity of phenolic-rich extracts has been extensively tested in various fields. Moreover, assessing the antimicrobial effect involves conducting in vitro assays against skin-associated pathogenic microorganisms, evaluating the inhibitory effect on bacterial and fungal growth to understand the ability of the compounds to safeguard the skin against infections. When focusing on cosmetics, it is evident that multiple types of infections can affect the skin, leading to dermocosmetic issues such as acne, a highly prevalent dermatological condition worldwide. Among the pivotal factors contributing to the development of this condition is the bacterial colonization by bacteria of the genus *Propionibacterium* and *Staphylococcus*, which subsequently triggers an inflammatory process [136]. Hence, phenolic compounds demonstrate efficacy not solely due to their antimicrobial capabilities, which will be elucidated below, but also owing to their previously mentioned anti-inflammatory attributes. Their antimicrobial potency arises from their capacity to inhibit the growth and proliferation of pathogenic microorganisms, which is of paramount importance in combating skin infections. For instance, in vitro studies have demonstrated the inhibitory effects of plant extracts rich in phenolic compounds, such as green tea extract (*Camellia sinensis*), against *P. acnes*, *P. granulosum*, *S. aureus*, or *S. epidermidis*—microorganisms implicated in the development of acne. The results of this study indicated a remarkable inhibition of 98% of bacterial growth at a concentration of 400 μg gallic acid equivalents mL^−1^ of the green tea extract [137]. Similar outcomes were achieved with an extract obtained from coffee silverskin, which exhibited significant antimicrobial activity against *S. aureus* and *S. epidermidis*, without cytotoxicity in skin cells such as fibroblasts and keratinocytes [128]. In the context of acne, various types of cinnamon extract have demonstrated activity against two bacteria known to cause acne, *P. acnes* and *S. epidermidis*. In this study, the authors attributed the antibacterial activity to phenolic compounds such as cinnamaldehyde and eugenol [138,139]. Pomegranate is another popular source of phenolic compounds. Pomegranate extract derived from peels has shown effectiveness in inhibiting several dermatophyte fungi, including *Trichophyton rubrum*, *Trichophyton mentagrophytes*, and *Microsporum canis* [140]. Furthermore, significant antibacterial activity was observed against *E. coli*, *S. aureus*, and *Listeria* [141]. Due to this potent antibacterial activity, coupled with its antioxidant activity that is primarily attributed to phenolic compounds, specifically punicalagin, pomegranate by-product extracts are being considered for various cosmetic applications.

Phenolic compounds, which are abundant in various plant sources, offer a multifaceted range of benefits for skin health. Their antioxidant, anti-inflammatory, and antimicrobial properties make them promising candidates for cosmetic applications. Further investigation into their mechanisms of action and clinical studies will likely unveil even more opportunities for these compounds in the skincare industry.

## 6. Ex Vivo Effects

Prior to, alternatively, and concurrently with in vivo studies, ex vivo assays utilizing models of human skin or epidermis, such as Franz cells, are crucial in validating the skin penetration and biological activity of bioactive ingredients, which, in this instance, are phenolic compounds formulated as emulgels, gels, creams, etc. [1,142]. The dermal stability, accessibility, and permeability of polyphenols are constrained by factors such as limited water solubility, reduced bioavailability, rapid metabolism, and systemic clearance [143]. Therefore, the penetration and vehicle dynamics of phenolic compounds have been widely studied as well as methods for preserving their bioactivity and their cosmetic potential regarding depigmenting, anti-collagenase, anti-inflammatory, antioxidant, elasticity, moisturizing activities, etc. [92,144,145]

Numerous studies have investigated various phenolic plant sources, vehicles, and cosmetic formulations’ efficacy ex vivo [142,146,147,148]. For instance, a study on the stratum corneum permeability of several phenolic compounds (chlorogenic acid, rutin, quercetin, (+)-catechin, and (–)-epicatechin) from an apple extract, in different vehicles, revealed that only the first three were capable of penetrating the skin’s outermost layer. Specifically, rutin could only penetrate skin layers when formulated in ointment and oleogel, two semi-solid bases, as opposed to when formulated in emulgels, gels, or emulsions [142].

Another case involves mushroom extracts, where, depending on the species of origin of the phenolic extract, different compounds are present, resulting in varying behavior in skin penetration. It was observed that the extract from *Gandoderma lucidum*, rich in protocatechuic and syringic acids, can penetrate the skin layers effectively, enhanced by the presence of triterpenoids in the extract, unlike the extract rich in cinnamic, p-hydroxybenzoic, and p-coumaric acids from *Pleurotus ostreatus* [146].

Despite the promising outcomes and the growing body of research on the activity, toxicity, and penetrability of phenolic compounds ex vivo, further studies are necessary to affirm their beneficial properties as well as to minimize reliance on animal models and to avoid extrapolating findings to compounds or products that may not be optimal in human models.

## 7. In Vivo Effects

Although in vitro and ex vivo studies have demonstrated the significant potential of plant extracts enriched in phenolic compounds for their application in cosmetic formulations, it is essential to ensure the reproducibility and non-toxicity of these results for users. In this way, as the use of natural ingredients has been associated with some adverse cases of skin allergies, conducting in vivo studies is still necessary to guarantee and evaluate their efficacy, determine the appropriate dosage, and address any potential toxicological concerns [149,150,151]. Nevertheless, the gradual transition to in vivo testing from quantitative in vitro and computational (in silico) approaches is a priority endorsed by REACH (Registration, Evaluation, Authorisation, and Restriction of Chemicals), in line with the 3Rs principle and the use of alternative methods to minimize animal testing in current regulatory practices and also considering that the Cosmetic Regulation (EC 1223/2009 https://eur-lex.europa.eu/legal-content/ES/ALL/?uri=CELEX%3A32009R1223, accessed on 14 May 2024) bans in vivo animal testing [151,152]. Therefore, advantageous cosmetics properties shown in vitro must be tested in vivo to confirm them [153].

While many plants enriched in phenolics have had their extracts characterized, only a few have been tested in vivo to determine their potential in cosmetics or pharmacosmetics in addition to in vitro studies. However, the ones that have been studied show promising results in these areas (Table 2). But no studies have been performed conferring a concrete phenol to a specific skin activity. Therefore, the activity has been related instead to an extract with a complex chemical composition. For example, extracts rich in caffeic acid, caffeoylquinic acids, quercetin, etc., including anthocyanins, exhibited anti-melanogenesis activity by powerfully inhibiting tyrosinase (IC50 = 12,48) and melanin synthesis in vitro [154]. Also, no irritation was shown when applied in guinea pigs (*n* = 33) and in female human skin (*n* = 50) during 24 h. In addition, skin-whitening, moisturizing, and erythema-reducing activities are attributed to them when applied as a 4% oil-in-water cream containing concentrated 4% ethanolic extract on the human male face for 8 weeks (*n* = 11) by inhibiting melanin production (measured with a mexameter), retaining water content (measured by corneometry), and restoring the ability of the lipid barrier to attract, retain, and redistribute water, thus maintaining the integrity and appearance of the skin [155].

Also, high phenolic content, including chlorogenic acid, caffeic acid, ferulic acid, and flavonoids, among others, such as that found in coffee extract has recently emerged as a potential extract in natural dermocosmetics. Certain fractions enriched in phenolic compounds, such as α-tocopherol, have been tested through topical application (5 mg/cm^2^) in female hairless mice (*n* = N.I. (not indicated) 24 h after UV irradiation, showing promising dermocosmetic effects, namely acting as antioxidants and providing photoprotective activities, which were performed by reducing the formation of epidermal lipid hydroperoxides, i.e., halving the glutathione content [156,157,158]. In the same way, a product was formulated with an extract rich in caffeic acid, chlorogenic acid, coumaric acid, cynarin, ferulic acid, luteolin naringenin, narirutin, and quinic acid, as provided by artichoke, at a concentration of 0.002% (*p*/*v*); this extract was shown to enhance skin roughness and elasticity. Over the course of a month-long application on female human sagging facial skin (*n* = 20), it demonstrated remarkable antioxidant, anti-inflammatory, and anti-aging properties, as it improved the wrinkle depth and skin elasticity properties. Furthermore, it acts as a protective agent for endothelial and lymphatic cells while also inhibiting vascular aging processes [61].

Extracts rich in caffeic acid, chlorogenic acid, cinnamic acid, coumaric acid, gallic acid, kaempferol, and sinapic acid, as contained in tomato, have resulted in pharmacosmetics interest with optimal in vivo results regarding anti-inflammatory and anti-allergy activities in intravenous and nutraceutical studies due to their great phenolics and carotenoids content [159]. For example, a 7% cream-based aqueous extract exhibited softening and moisturizing effects on human arm skin (*n* = 40) 48 h after the application and on feet skin 24 h after the application, showing promise in the emerging field of tomato dermocosmetics [160]. However, further investigations are necessary to explore the molecular antioxidant mechanisms or determine the preservation of these extracts as dermocosmetic formulations, given that the majority of investigations are orientated to gastric or medical treatments [81].

Extracts rich in phenolics like flavonoids (quercetin), phenolics acids (hydroxybenzoic and hydroxycinnamic acids), and stilbenes, such as those contained in grape [161,162], have emerged as one of the most extensively studied extracts in this area. In fact, there is substantial research about formulations where grape (*V. vinifera*) phenolics are identified as potential active ingredients in cosmetics. Some studies demonstrate a sunscreen protective effect of this extract in both mice and human, thereby enhancing the value of cosmetic formulations. Before UV-B radiation, a hydroethanolic extract prevented radiation damage by reducing the levels of pro-inflammatory cytokines and sunburn cells 24 h after a single application in female hairless mice (*n* = 40). These results respond to a significant decrease in pro-inflammatory cytokines level and antioxidant activity [161]. Furthermore, a 10% grape extract formulated in an oil-in-water emulsion with UV filters like butylmethoxydibenzoyl methane was tested on human skin (*n* = 60) for 6 weeks. It presented significant photoprotective effectiveness in reducing erythema formation by 21% after UV exposure compared to a sunscreen formulation alone [163]. Other studies in humans reveal the antioxidant properties of grape extract formulation, which improved various parameters, including radiant glow, smoothness, hydration, texture, and softness, primarily due to a decrease in reactive oxygen species (ROS) in the stratum corneum of photoaged skin (through a radical scavenging mechanism) after 4 weeks of twice daily application in female human forearm skin (*n* = 60) [164]. Notably, the “Muscat Hamburg” variety demonstrated great potential in improving skin elasticity and inhibiting erythema and hyper-pigmentation. This effect was observed when formulated as a 2% water-in-oil emulsion and tested on male human cheeks (*n* = 110) over an 8-week period during winter. The effects were attributed to the tyrosinase-inhibitory activity of resveratrol (one of the most characterized and utilized stilbenes in cosmetology) and the enzymatic-inhibitory activity (acting as an anti-collagenase) of grape phenols, which prevent skin aging and deterioration [111,165]. In fact, a specific trans-resveratrol enriched extract from grapes, formulated as a 0.1% water-in-oil cream, revealed significant anti-aging effects and improvements in skin parameters (measured by colorimetry, elastometry, and corneometry) when applied over a month to aged female human skin (*n* = 8). The results were further enhanced when formulated with β-cyclodextrin, which increased the efficacy of resveratrol action [111,166].

Flavonones and phenolic acids such as chlorogenic acid, cinnamic acid, ellagic acid, hesperidin, kaempferol, luteolin, naringenin, naringin, narirutin, and quercetin, mainly located in the peels of citrus fruits, also represent innovative candidates in this field. These compounds have been extensively investigated in the field of biomedicine [167]. However, despite numerous in vitro findings suggesting the beneficial cosmetic effects of phenolic compounds obtained from citrus, particularly in pharmacosmetics, comprehensive in vivo studies are lacking [168]. These extracts, including essential oils, have been evaluated for their anti-inflammatory effects [169,170]. Citrus extract showed an effective counteracting activity against the effects of UV exposure and photo-aging, improving skin parameters such as elasticity or moisturizing and decreasing lipid peroxidation and erythema generation [171]. A 3% mandarin ethanolic extract exhibited anti-inflammatory properties when applied for 36 days in an atopic dermatitis female hairless mouse model. This application led to a reduction in redness, hyperkeratosis, and wrinkles, showcasing its potential utility in anti-atopic formulations [172]. All these effects could be attributed to phenolic compounds such as naringenin and hesperidin since some in vivo studies using these individual compounds demonstrated photoprotective properties when they deeply penetrate the skin, as observed in tests on humans after UV-induced erythema [173,174].

The phenolic extracts with a high content in chlorogenic acid, caffeic acid, coumaric acid, ferulic acid, kaempferol, quercetin, and sinapic acid, as obtained from brassicas, have been investigated in vitro, but the in vivo mechanisms when they are included in cosmetic formulation have rarely studied [175,176,177]. In this way, extracts from *Brassica oleracea* var. *capitata* f. *rubra* (red cabbage), when formulated as an ethosomal gel (2% Carbopol gel), demonstrated significant antioxidant activity with improvements in smoothness, reduced wrinkles, minimized facial pore size, enhanced skin hydration (measured by corneometry), increased elasticity, regulated sebum production, decreased erythema, and balanced melanin levels observed when applied to female human cheeks [178]. These findings led to the proposition of red cabbage extract as a potential active for dermatitis or acne vulgaris [179]. Furthermore, chlorogenic- and quercetin-enriched ethanolic extract obtained from *Brassica nigra* (black mustard) demonstrated significant anti-inflammatory activity. However, topic administration tests of specific dermocosmetics have not been conducted in either chlorogenic- or quercetin-enriched extract [180]. Collectively, these results indicate promising potential for formulating dermocosmetics based on brassicas. Nonetheless, comprehensive in vivo investigations including topical tests are required to fully understand and harness their benefits.

Also, the vegetal extracts rich in coumaric acid, ferulic acid, vanillic acid, and quercetin from plants such as *Allium cepa* (onion) and *Allium sativa* (garlic) or those rich in caffeic acid, chlorogenic acid, cinnamic acid, coumaric acid, ferulic acid, gallic acid, vanillic acid, luteolin, myricetin, naringenin, catechin, and rutin, such as from *Agaricus bisporus* (mushroom), have presented high antioxidant effects by radical scavenger or lipid peroxidation inhibition mechanisms, showing moisturizing and anti-inflammatory effects. However, again, these extracts have not been tested in vivo [146,181,182,183,184].
ijms-25-05884-t002_Table 2Table 2Summary of in vivo studies or tests related to dermocosmetics of plant species enriched in phenolics.Plant SpecieFormulationCohortIn Vivo ResultsReferenceArtichokeEthanolic extract formulated in cream (0.002%)Female human with sagging faceImprovement of endothelial cell integrity by enhancement of skin roughness and elasticity[61]*Brassica*Ethosomal Carbopol 2% gelFemale humanStrong antioxidant activity with remarkable dermocosmetic benefits for skin.[179]*Citrus*Aqueous/Aqueous–ethanolic solution extractHealthy human with UVB-induced skin erythemaAntioxidant by radical scavenger activity: skin photoprotection[174]ROC^TM^Human forearmPhotoprotective, anti-erythematic, and antiaging activities[184]Ethanolic extractFemale SKH-1 hairless mice.Anti-inflammatory activity, useful as an anti-atopic agent.[172]GrapeHydroethanolic extract (4 mg/40 μL/cm^2^)SKH-1 hairless mice.Photoprotective activity (against UV-B skin damage)[161]Oil-in-water solution (10%)Healthy humanPhotoprotective activity, synergic with sunscreen chemicals formulation[163]Antioxidant and anti-photoaging activities.[164]Sarmentine (1%) creamHumanAnti-aging, moisturizing, and skin-whitening activities[165]β-ciclodextrine formulated with solution (0.1%)HumanAnti-aging activity[166]TomatoGlycerinated formulation (3%)HumanMoisturizing activity[160]


## 8. Concluding Remarks

Phenolic compounds that are part of the secondary metabolism in plant tissues are gaining considerable attention for their potent antioxidant and anti-inflammatory properties, conferring anti-ageing, photoprotective, and antimicrobial properties in cosmetic products. Therefore, this could be a high-value usage for several by-products that are produced worldwide. However, as has been reviewed in this work, there are some issues, from the harvest to the in vivo assays, that must be considered to reach an effective cosmetic formula. At first instance, one should bear in mind that plant material by-products should be considered as primary material, and the chemical composition should be maintained as material for food. Therefore, harvest and storage should be carried out in the same way, or all the bioactive compounds will be lost. Secondly, the extraction process (technology, solvent, pressure, time, etc.) should be carefully chosen according to the specific phenolic compounds of interest. It is on this topic that there is a great lack of knowledge. The actions of phenolic compounds on skin can be summarized as antioxidant, anti-inflammatory, and antimicrobial, which have been extensively studied in vitro. The strategies to undertake in vitro assays serve as a crucial tool in elucidating the properties and in vivo mechanisms of action of these compounds, but in vivo studies are very scarce and need to be implemented. Therefore, the fact that plant phenolics may be efficient in the treatment of both serious life-threatening dermal diseases and minor skin problems is of high value due to the natural source of these compounds. In addition, the recovery of these compounds has a positive impact on mitigating climate change due to the circular economy. Furthermore, further investigation of the innate qualities and remarkable efficacy of phenolic compounds from crop by-products offers promising prospects for the development of innovative topical formulations and dressings that could potentially replace existing therapeutic applications.

## Figures and Tables

**Figure 1 ijms-25-05884-f001:**
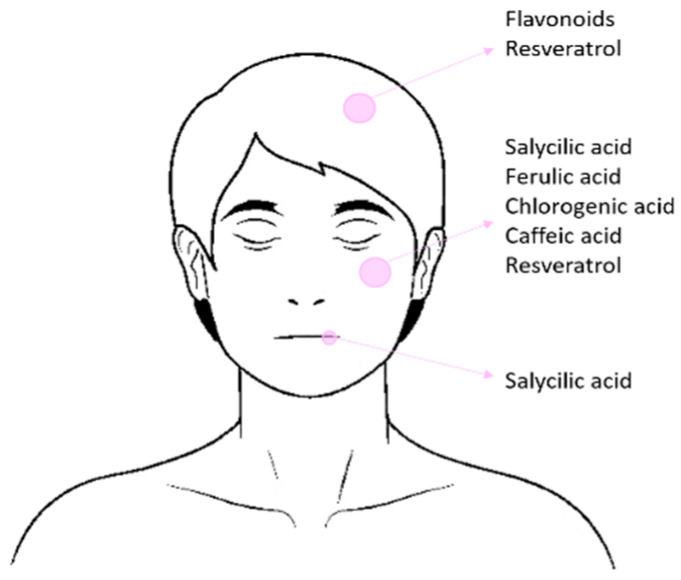
Phenolic compounds present in the formulations of products marketed in the EU.

**Figure 2 ijms-25-05884-f002:**
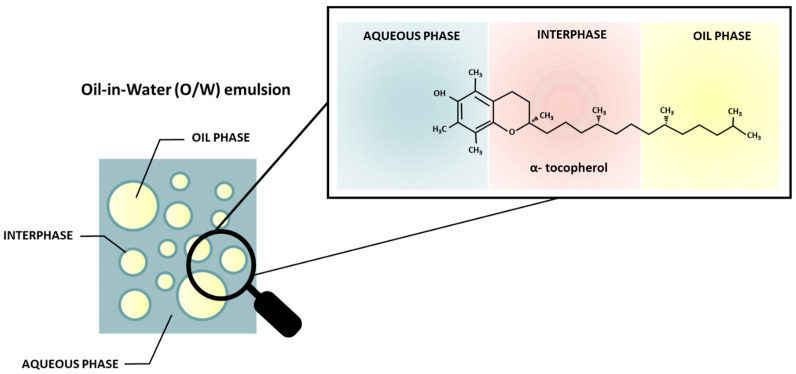
Scheme of the different phases present in an oil-in-water emulsion and an example of a phenolic molecule distribution.

**Figure 3 ijms-25-05884-f003:**
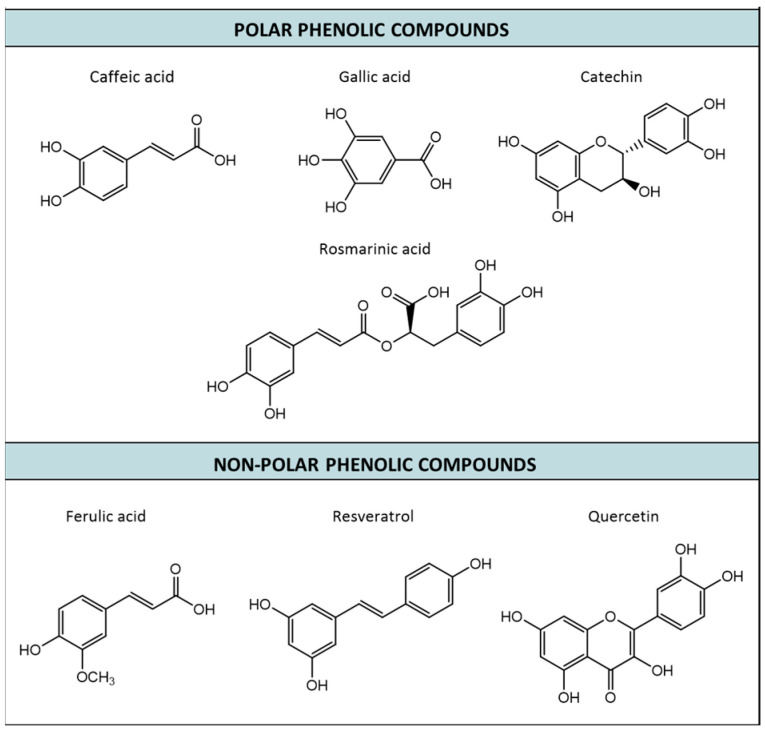
Structure of main polar (hydroxytyrosol, caffeic acid, gallic acid, catechin, and rosmarinic acid) and non-polar (salicylic acid, ferulic acid, resveratrol, quercetin, curcumin, and α-tocopherol) phenolic compounds of interest in cosmetics.

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
