# Peer review of "Exploring Phenolic Compounds in Crop By-Products for Cosmetic Efficacy"

_ijms, 2024, doi:10.3390/ijms25115884_

Round 1

Reviewer 1 Report

Comments and Suggestions for Authors

The review touches on an interesting and extremely relevant topic with significant implications for the green economy and the reuse of waste. However, it seems that it lacks a coherent organization and a clear focus that guides its content.

Firstly, the review discusses waste from 5-6 different crops (such as oil production, wine production, tomatoes, mushrooms, onions, and lemons), albeit to varying degrees. Perhaps, by addressing too many different types of food, which can result in completely different yields in terms of polyphenols, it may not be the best choice. The varying percentages of phenols or polyphenols contained in these foods, as well as their heterogeneity, make it difficult to organize a cohesive and fluid discussion that does not seem like a mere collection of articles.

Secondly, in my opinion, the review covers too broad a spectrum of topics - ranging from extraction methods to in vivo testing, from brief structural descriptions to cream formulations. As a result, none of the topics seem to receive adequate attention. For instance, readers interested in the structural aspect of the research would likely desire a more in-depth examination of compound structures or their analogs, spanning a diverse array from quercetin derivatives to tocopherols to caffeic acid. Conversely, individuals seeking insights into optimal extraction techniques for these waste materials may struggle due to the inherent variability among samples.

Furthermore, there is a notable absence of references to recent reviews that delve deeply into specific themes within this field. Some are here listed:

·       Cosmetics | Free Full-Text | Olea europea and By-Products: Extraction Methods and Cosmetic Applications (mdpi.com)

·       Feeding the skin: A new trend in food and cosmetics convergence - ScienceDirect

·       Cosmetics | Free Full-Text | Coffee Silverskin: A Review on Potential Cosmetic Applications (mdpi.com)

·       Natural products in cosmetics | Natural Products and Bioprospecting (springer.com)

MINOR:

R412: The term "acid" has been repeated.

R407: alpha – α

R 414: “at a concentration of 0.002%” p/p, v/v?

In some instances, "antinflammatory" and "antioxidant" are highlighted in the text, while in others they are not.

Comments on the Quality of English Language

acceptable

Author Response

REVIEWER 1

Comments and Suggestions

The review touches on an interesting and extremely relevant topic with significant implications for the green economy and the reuse of waste. However, it seems that it lacks a coherent organization and a clear focus that guides its content.

Firstly, the review discusses waste from 5-6 different crops (such as oil production, wine production, tomatoes, mushrooms, onions, and lemons), albeit to varying degrees. Perhaps, by addressing too many different types of food, which can result in completely different yields in terms of polyphenols, it may not be the best choice. The varying percentages of phenols or polyphenols contained in these foods, as well as their heterogeneity, make it difficult to organize a cohesive and fluid discussion that does not seem like a mere collection of articles.

In this review, we establish the relationship of phenol-rich natural extracts obtained from plant by product as functional ingredients in the cosmetic sector. This consideration includes typical Mediterranean crops and highlights the gaps in knowledge that need to be addressed for the integration of these extracts into the cosmetic sector. The review only has taken into account fresh food as artichoke, broccoli, cauliflower, citrus, grape, tomato, onion and mushrooms. We agree with the referee about the heterogeneity of the polyphenol composition in this materials, but this is one of the main points to remark. The cosmetic sector need to differentiate each individual compound to generate the specific target.

Secondly, in my opinion, the review covers too broad a spectrum of topics - ranging from extraction methods to in vivo testing, from brief structural descriptions to cream formulations. As a result, none of the topics seem to receive adequate attention. For instance, readers interested in the structural aspect of the research would likely desire a more in-depth examination of compound structures or their analogs, spanning a diverse array from quercetin derivatives to tocopherols to caffeic acid. Conversely, individuals seeking insights into optimal extraction techniques for these waste materials may struggle due to the inherent variability among samples.

As part of the secondary metabolism in plant tissues, phenolic compounds are getting considerable attention for their potent antioxidant and anti-inflammatory properties, conferring anti-ageing, photoprotective, and antimicrobial properties in cosmetic products. However, we wanted to remark in this work that there are some issues, from the harvest to the in vivo assays, that must be considered to reach an effective cosmetic formula. Probably our work does not deeply get inside each of the subject’s deal, but report the gasps in the knowledge in each of this subjects.

Furthermore, there is a notable absence of references to recent reviews that delve deeply into specific themes within this field. Some are here listed:

  • Feeding the skin: A new trend in food and cosmetics convergence – ScienceDirect
  • Cosmetics | Free Full-Text | Coffee Silverskin: A Review on Potential Cosmetic Applications (mdpi.com)
  • Natural products in cosmetics | Natural Products and Bioprospecting (springer.com)

New references have been considered and added to the review.

 MINOR :

R412: The term "acid" has been repeated.

R407: alpha – α

R 414: “at a concentration of 0.002%” p/p, v/v?

In some instances, "antinflammatory" and "antioxidant" are highlighted in the text, while in others they are not.

All mistakes have been corrected.

Reviewer 2 Report

Comments and Suggestions for Authors

The review of this team of authors is related to the importance of using phenolic compounds that are found in reaction byproducts. Due to their antioxidant and anti-inflammatory properties, they can be useful in cosmetology as anti-aging, photoprotective and antimicrobial agents. Thus, further study of the innate qualities and remarkable effectiveness of plant byproducts of phenolic compounds opens up promising prospects for the development of innovative drugs for topical use of formulations and dressings that can potentially replace existing therapeutic agents. Despite its impressive scope of review and the full use of literary sources. There are comments to the review that do not reduce the impression, but should be discussed before publication in Int. J. Mol. Sci.

1) Provide several links to research related to the use of phenolic compounds (10.1016/j.indcrop.2015.12.016, 10.1111/ijfs.17056, 10.1080/10408398.2016.1224805, 10.1111/ijfs.17056, 10.1002/pca.2720)

2) If the use of vegetables such as potatoes, dill, legumes? Add the data to table 1.

3) Polar phenolic and non-polar compounds starting from page 4 are better combined in structure. This is how readers will clearly perceive it.

4) To study the effects in vitro and in vivo, it is worth mentioning the effects of ex vivo. Dedicate a small section to this part.

5) Provide all literary sources with DOI information. Follow the general rules of registration. Example, Journal name in italics, year in bold, volume in italics.

Author Response

Comments and Suggestions

The review of this team of authors is related to the importance of using phenolic compounds that are found in reaction byproducts. Due to their antioxidant and anti-inflammatory properties, they can be useful in cosmetology as anti-aging, photoprotective and antimicrobial agents. Thus, further study of the innate qualities and remarkable effectiveness of plant byproducts of phenolic compounds opens up promising prospects for the development of innovative drugs for topical use of formulations and dressings that can potentially replace existing therapeutic agents. Despite its impressive scope of review and the full use of literary sources. There are comments to the review that do not reduce the impression, but should be discussed before publication in Int. J. Mol. Sci.

1) Provide several links to research related to the use of phenolic compounds (10.1016/j.indcrop.2015.12.016, 10.1111/ijfs.17056, 10.1080/10408398.2016.1224805, 10.1111/ijfs.17056, 10.1002/pca.2720)

Thanks, the new articles have been cited.

2) If the use of vegetables such as potatoes, dill, legumes? Add the data to table 1.

Although there are a few studies on phenolic compounds relevant to cosmetics present in such crops, such as legumes https://doi.org/10.3390/cosmetics7040084), dill (https://doi.org/10.1111/ics.12644), or potato (https://doi.org/10.1016/j.jksus.2021.101532 or https://doi.org/10.3390/antiox11071401).  However, as potatoes, dill, and legumes have specific growth habits and environmental requirements that may differ significantly from the plants taken into account in our study, we prefer to concentrate into the cited vegetables. Also, while potatoes, dill, and legumes have their own importance in agriculture and horticulture, using the extracts of by-products in cosmetic could unwanted variables and complications.

3) Polar phenolic and non-polar compounds starting from page 4 are better combined in structure. This is how readers will clearly perceive it.

The change has been included

4) To study the effects in vitro and in vivo, it is worth mentioning the effects of ex vivo. Dedicate a small section to this part.

Small section added.

5) Provide all literary sources with DOI information. Follow the general rules of registration. Example, Journal name in italics, year in bold, volume in italics.

DOI information have been added to references 33, 124, 160, 196, 197, 199, 201, 204, 207 and 210.

References number 50, 112, 113, 180, 183, 185, 202, 208 and 211 have no DOI information associated.

The rest of mistakes have been reviewed and corrected.

Reviewer 3 Report

Comments and Suggestions for Authors

This review article on the cosmetic use of phenolic compounds in crop by-products is informative and important in exploring proper uses of otherwise wasted plant materials. This review gives us a comprehensive overview on the composition in phenolic compounds of by-products, extraction processes and stability of compounds, and cosmetic formulation, in vitro effects and in vivo effects. However, this review can be improved by addressing the following points. 

1)    This review will appear more interesting to chemically oriented readers by showing structure of typical phenolic compounds of interest. They may include: caffeic acid, gallic acid, rosmaric acid, hydroxytyrosol, alfa-tocophenol, curcumin, quercetin, salicylic acid, ferulic acid, and resveratrol (those are title compounds in the section 4).

2)    There are many sentences that are incorrect or vague in the use of English. Some examples are: Line 16 (extract method), Line 23 (to a proper storage plant material), Line 36 (phenolic compounds structure), Line 5 (polar o non-polar), Line 52 and 53 (SC-CO2), Line 58 ([82]broccoli [83]), Line 59 (for Artichoke), Line 142 (3,4-dihydroxy…), Line 148 (o by), Line 185 (to a -OH ring), Line 206 (3’-4’dihydroxy), Line 224 (In this way, emonstrating…), Line 281(tested and resulting in protect keratinocytes…), 

3)    “-O-“ should be “-O-“. “ortho-“ should be “ortho-”. 

4)    “p-OH-benzoic acid” should be “4-hydroxybenzoic acid”. There are several of this including those in Table 1.

5)    Give full names for FRAP and ORAC (Line 261).

Comments on the Quality of English Language

I suggest that the manuscript be edited by α native speaker. 

Author Response

REVIEWER 3

Comments and Suggestions

1) This review will appear more interesting to chemically oriented readers by showing structure of typical phenolic compounds of interest. They may include: caffeic acid, gallic acid, rosmaric acid, hydroxytyrosol, alfa-tocophenol, curcumin, quercetin, salicylic acid, ferulic acid, and resveratrol (those are title compounds in the section 4).

They have been added.

2) There are many sentences that are incorrect or vague in the use of English. Some examples are: Line 16 (extract method), Line 23 (to a proper storage plant material), Line 36 (phenolic compounds structure), Line 5 (polar o non-polar), Line 52 and 53 (SC-CO2), Line 58 ([82]broccoli [83]), Line 59 (for Artichoke), Line 142 (3,4-dihydroxy…), Line 148 (o by), Line 185 (to a -OH ring), Line 206 (3’-4’dihydroxy), Line 224 (In this way, emonstrating…), Line 281(tested and resulting in protect keratinocytes…).

The sentences have been corrected according to the suggestion of the referee.

 3)”-O-“ should be “-O-“. “ortho-“ should be “ortho-”. 

It has been corrected

4)”p-OH-benzoic acid” should be “4-hydroxybenzoic acid”. There are several of this including those in Table 1.

It have been corrected.

5) Give full names for FRAP and ORAC (Line 261).

The full names have been provided

Comments on the Quality of English Language

I suggest that the manuscript be edited by α native speaker. 

The review has been revised very carefully and with an editing program. If still the quality is not acceptable, we will send it to a native speaker.

Round 2

Reviewer 1 Report

Comments and Suggestions for Authors

The authors did not try to re-organize the review or reply to my major concerns. To accept the review please consider focusing better on the criticisms in by-product collection, type, time, and the recovery of important secondary metabolites. Moreover, try to suggest useful guidelines in this sense.

Comments on the Quality of English Language

The English sounds good, but some minor errors in the text should be corrected.

Author Response

The authors did not try to re-organize the review or reply to my major concerns. To accept the review please consider focusing better on the criticisms in by-product collection, type, time, and the recovery of important secondary metabolites. Moreover, try to suggest useful guidelines in this sense.

Following the referee suggestion (we have to recognize that the manuscript has been improved now) we have now introduced substantial changes in the manuscript reorganizing plant material and specific compounds. In this way, a paragraph has been included according to by-product collection, type, time, and the recovery of important secondary metabolites, since the heterogeneity has been associated to that mater. Also, a column with the described concentration in plant by-product material has been added to Table 1. We have been centred in the Mediterranean crops removing any reference to other plants and highlighted the cosmetic referenced effect of some of the phenolics.

Round 3

Reviewer 1 Report

Comments and Suggestions for Authors

The author re-organize the review and now is well structured.

Comments on the Quality of English Language

English requires checking for minor errors